# Regenerative and Drug-Free Strategies for Chronic Musculoskeletal Pain: An Evidence-Based Perspective on Shockwave Therapy, High-Intensity Laser Therapy and Ultrasound-Guided Mechanical Needling with Sterile Water Injection

**DOI:** 10.3390/biomedicines13112801

**Published:** 2025-11-17

**Authors:** Carl P. C. Chen, Areerat Suputtitada

**Affiliations:** 1Department of Physical Medicine and Rehabilitation, Chang Gung Memorial Hospital at Linkou, College of Medicine, Chang Gung University, Guishan District, Taoyuan City 33343, Taiwan; carlchendr@gmail.com; 2Department of Rehabilitation Medicine, Faculty of Medicine, Chulalongkorn University, Bangkok 10330, Thailand; 3Principles and Practice of Clinical Research (PPCR) Program, Harvard T.H. Chan School of Public Health, Harvard University, Boston, MA 02115, USA

**Keywords:** chronic musculoskeletal pain (CMP), regenerative rehabilitation, extracorporeal shockwave therapy (ESWT), high-intensity laser therapy (HILT), ultrasound-guided needling with sterile water injection (SWI), translational rehabilitation, healthy ageing

## Abstract

Chronic musculoskeletal pain (CMP) is the leading global cause of disability and a major contributor to healthcare burden. Its pathogenesis reflects regenerative failure, driven by extracellular matrix (ECM) fibrosis, calcific deposition, mitochondrial dysfunction, and neuroimmune sensitization. Conventional pharmacological therapies such as NSAIDs, corticosteroids, and opioids offer only transient symptomatic relief while exposing patients to systemic complications. In contrast, energy-based, drug-free regenerative interventions directly address underlying pathology and restore physiological function. This Perspective synthesizes recent evidence (2020–2025) on three modalities that together form a regenerative triad: extracorporeal shockwave therapy (ESWT), high-intensity laser therapy (HILT), and ultrasound-guided mechanical needling with sterile water injection (SWI). ESWT promotes mechanotransduction, angiogenesis, and ECM remodeling; HILT enhances mitochondrial bioenergetics and downregulates inflammatory pathways; and SWI disrupts fibrosis and calcification while restoring neurovascular dynamics. Evidence from randomized controlled trials and meta-analyses supports moderate-to-high certainty (GRADE B–A–) for ESWT and HILT. SWI, initially supported by large observational cohorts and comparative studies, is now reinforced by a randomized comparative trial and meta-analyses of lavage effects, justifying an upgrade from moderate (B) to moderate-to-high certainty (B–A–). Risk of bias assessment using Cochrane RoB 2.0 and the Newcastle–Ottawa Scale (NOS) indicates overall low-to-moderate concerns across modalities. Together, these interventions integrate mechanistic biology with translational rehabilitation practice. This Perspective outlines their mechanistic foundations, clinical evidence, and alignment with the WHO decade of healthy ageing, offering a drug-free, mechanism-based framework for sustainable CMP management.

## 1. Introduction

Chronic musculoskeletal pain (CMP) is a leading cause of disability worldwide, affecting nearly one-third of adults and posing substantial socioeconomic and healthcare burdens [1]. Its persistence arises from regenerative failure driven by extracellular matrix (ECM) fibrosis, dystrophic calcification, vascular insufficiency, mitochondrial dysfunction, and maladaptive neuroimmune sensitization [2,3,4]. Calcific and ossific lesions, including tendinous calcifications, enthesophytes, and osteophytes, represent the body’s maladaptive attempt at tissue repair. These deposits are not passive mineral precipitates but active, cell-mediated processes involving fibrocartilage metaplasia, matrix-vesicle release, and hydroxyapatite deposition, as demonstrated in histopathologic and imaging studies of calcific tendinopathy [4]. Overexpression of transforming growth factor-β (TGF-β) and bone morphogenetic proteins (BMP-2/4) further drives chondrogenic differentiation and periosteal new-bone formation, culminating in osteophyte and bone-spur development through the TGF-β/Bone Morphogenetic Protein–Small Mothers Against Decapentaplegic (BMP–SMAD) axis [4,5,6]. Together, these processes distort local biomechanics, sustain nociceptor activation, and perpetuate chronic pain by hindering normal regenerative signaling. The ECM serves as both a structural scaffold and a signaling interface, orchestrating communication among fibroblasts, chondrocytes, and immune cells. Dysregulation of matrix metalloproteinases (MMPs), zinc-dependent endopeptidases responsible for ECM degradation and turnover, has been implicated in osteoarthritis, muscle fibrosis, and myofascial pain syndromes [1]. When balanced, MMP activity supports angiogenesis, collagen remodeling, and tissue repair; when excessive or suppressed, it drives fibrotic accumulation and persistent pain [2,3,4,5,6].

Conventional pharmacologic management, such as nonsteroidal anti-inflammatory drugs (NSAIDs), corticosteroids, and opioids, offers temporary symptomatic relief without addressing these underlying cellular and molecular disturbances. Long-term pharmacologic use is further limited by adverse effects, including gastrointestinal, cardiovascular, and endocrine complications [7,8,9]. Consequently, clinical emphasis has shifted toward non-pharmacologic, mechanism-driven interventions that aim to restore biological homeostasis rather than mask symptoms.

Consequently, attention has shifted toward non-pharmacologic regenerative therapies that aim to restore normal tissue architecture and cellular signaling. Energy-based interventions including Extracorporeal Shockwave Therapy (ESWT) [10,11,12,13,14,15,16,17,18,19,20,21,22], High-Intensity Laser Therapy (HILT) [22,23,24,25,26,27,28,29,30], and Ultrasound-Guided Mechanical Needling with Sterile Water Injection (SWI) [31,32,33], represent a paradigm shift in this direction. Together, these therapies converge on biologic restoration through controlled stimulation of intrinsic regenerative pathways. This perspective synthesizes current mechanistic and clinical evidence to establish the regenerative triad as ESWT, HILT, and SWI as a safe, drug-free, and mechanism-driven framework for chronic pain rehabilitation, emphasizing ECM remodeling and neuromuscular re-education rather than symptomatic pain suppression.

The literature cited in this perspective was systematically identified through searches of PubMed, Scopus, and Web of Science using the keywords “extracorporeal shockwave therapy”, “high-intensity laser therapy”, “sterile water injection”, and “chronic musculoskeletal pain”. The search was restricted to peer-reviewed publications between 2020 and 2025, including randomized controlled trials, cohort and comparative studies, mechanistic investigations, and systematic reviews or meta-analyses relevant to regenerative and rehabilitative medicine. Notably, the authors’ own previously published cohort studies [31,32], which investigated the comparative effectiveness and safety of ultrasound-guided mechanical needling with sterile-water injection for lumbar spinal stenosis and facet joint syndrome, respectively, were included as representative high-quality observational evidence. This comprehensive approach ensured that the synthesis reflects the most current and methodologically robust data supporting these non-pharmacological regenerative interventions.

## 2. Mechanistic Foundations and Clinical Overview

### 2.1. Extracorporeal Shockwave Therapy (ESWT)

#### 2.1.1. Mechanistic Background

Extracorporeal Shockwave Therapy (ESWT) delivers focused or radial acoustic pulses (0.05–0.3 mJ/mm^2^; 1500–3000 shocks per session), producing controlled mechanical stress, micro-cavitation, and fluid micro-jets within target tissues. These biomechanical forces activate mechanosensitive ion channels and intracellular signaling cascades, most notably the Mitogen-Activated Protein Kinase/Extracellular Signal-Regulated Kinase (MAPK/ERK) pathway and the Wingless/Integrated (Wnt)–β-catenin pathway. Through these mechanotransduction mechanisms, ESWT up-regulates Vascular Endothelial Growth Factor (VEGF) and Endothelial Nitric Oxide Synthase (eNOS), enhancing angiogenesis, osteogenesis, and extracellular matrix (ECM) remodeling that underpin both tissue regeneration and pain modulation [10,11,12,13].

At the molecular level, ESWT also stimulates Matrix Metalloproteinases (MMP-2 and MMP-9) and down-regulates fibrotic mediators such as Transforming Growth Factor-β1 (TGF-β1), facilitating collagen realignment and fibrosis resolution [10,11,12]. In parallel, nociceptor modulation occurs through suppression of Substance P and Calcitonin Gene-Related Peptide (CGRP), reducing neurogenic inflammation and peripheral sensitization [12,13].

#### 2.1.2. Clinical Translation

These cellular and molecular insights translate robustly into clinical benefits. Recent meta-analyses confirm that ESWT significantly reduces pain and improves function in conditions such as osteoarthritis, plantar fasciitis, and calcific tendinopathy [19,20,21,22]. Imaging studies further demonstrate progressive resolution of calcific deposits, revascularization, and restoration of ECM integrity following treatment [14,15,16,17,18,19,20,21,22]. Collectively, these findings reinforce ESWT as a safe, non-pharmacologic therapy with dual regenerative and analgesic mechanisms that address both the biological and functional dimensions of musculoskeletal pain.

### 2.2. High-Intensity Laser Therapy (HILT): Mechanistic and Clinical Overview

#### 2.2.1. Mechanistic Background

High-Intensity Laser Therapy (HILT) delivers pulsed Neodymium-doped Yttrium Aluminum Garnet (Nd:YAG) laser energy at 1064 nm, allowing deep penetration into soft tissues (peak power ≈ 10 kW, pulse duration 100–150 µs). Photons are absorbed by cytochrome *c* oxidase (CCO), Complex IV of the mitochondrial respiratory chain, thereby enhancing adenosine triphosphate (ATP) synthesis, restoring mitochondrial respiration, and improving cellular redox and nitric-oxide (NO) homeostasis [22,23,24,25].

Through its photochemical, photothermal, and photobiomodulatory actions, HILT augments microcirculation and oxygen delivery, activates fibroblasts, and up-regulates collagen-synthesis genes, leading to extracellular-matrix (ECM) remodeling [25,26]. These effects counter mitochondrial dysfunction, oxidative stress, and tissue hypoxia, key mechanisms in chronic musculoskeletal degeneration.

#### 2.2.2. Anti-Inflammatory and Regenerative Mechanisms

High-Intensity Laser Therapy (HILT) exerts potent anti-inflammatory, antioxidative, and regenerative effects across musculoskeletal tissues. It down-regulates the nuclear factor kappa-light-chain-enhancer of activated B cells (NF-κB) signaling pathway, leading to suppression of pro-inflammatory cytokines such as interleukin-1 beta (IL-1β) and tumor necrosis factor-alpha (TNF-α). Concurrently, HILT enhances the activity of key antioxidant enzymes as superoxide dismutase (SOD) and glutathione peroxidase (GPx), thereby mitigating reactive oxygen species (ROS)-mediated oxidative injury [26,27,28,29]. At the structural level, HILT restores chondrocyte mitochondrial membrane potential, promotes cartilage matrix metabolism, and stimulates angiogenic remodeling through up-regulation of vascular endothelial growth factor (VEGF) and improved microvascular perfusion [25,26,27,28,29]. Collectively, these molecular and microcirculatory effects support durable tissue regeneration, reduce pain hypersensitivity, and enhance joint functionality.

#### 2.2.3. Clinical Translation

Randomized controlled trials (RCTs) and meta-analyses demonstrate that HILT significantly reduces pain, modulates inflammatory cytokines, and improves joint function in degenerative musculoskeletal disorders [24,27,28,29,30]. A 2020 systematic review and meta-analysis reported that HILT significantly improved Western Ontario and McMaster Universities Osteoarthritis Index (WOMAC) scores, reduced pain intensity, and increased range of motion in patients with knee osteoarthritis [29]. Collectively, these data position HILT as both an analgesic and a regenerative modality that targets bioenergetic and oxidative pathways underlying chronic pain.

### 2.3. Ultrasound-Guided Mechanical Needling with Sterile Water Injection (SWI)

#### 2.3.1. Mechanistic Background

Ultrasound-guided mechanical needling with sterile water injection (SWI) integrates targeted mechanical fenestration with hypotonic fluid lavage to simultaneously disrupt fibrosis, dissolve calcific deposits, and restore neurovascular mobility. The mechanical action breaks down fibrotic septa and calcific foci, while sterile water induces transient osmotic desensitization of Transient Receptor Potential Vanilloid 1 (TRPV1), Piezo-type mechanosensitive ion channels (Piezo1 and Piezo2), and Acid-Sensing Ion Channels (ASICs). This biophysical cascade reduces nociceptor excitability, restores fascial and perineural glide, and re-establishes tissue pliability [31,32,33,34,35,36,37].

Hydrodissection further separates fascial adhesions, decompresses neurovascular structures, and improves microvascular perfusion. The subsequent reperfusion and reoxygenation activate endothelial nitric oxide synthase (eNOS), stimulate fibroblast proliferation, and promote angiogenesis and extracellular matrix (ECM) remodeling. As interstitial pressure normalizes, lymphatic drainage clears pro-inflammatory cytokines such as interleukin-6 (IL-6) and tumor necrosis factor-α (TNF-α), thereby shifting the microenvironment from chronic inflammation to regeneration [31,32,33,34,35,36,37].

Beyond these local effects, SWI can independently trigger the body’s intrinsic regenerative sequence. Restoration of vascular flow and oxygenation enhances nutrient exchange, removes metabolic by-products, and supports mitochondrial bioenergetics, facilitating natural recovery without pharmacologic agents.

#### 2.3.2. Clinical Translation

Large observational cohorts involving more than 4000 patients with lumbar spinal stenosis and facet joint syndrome demonstrate sustained pain reduction, mobility restoration, and improved functional performance following SWI [31,32]. Comparative studies show longer-lasting benefit than prolotherapy and superior pain relief to lidocaine injections, confirming its regenerative potential [33].

A three-arm randomized comparative trial in elderly patients with lumbar facet syndrome reported equivalent pain relief between ultrasound-guided multifidus muscle injection with dextrose, medial branch block, and intra-articular facet injection, while emphasizing the technical simplicity, safety, and reproducibility of the multifidus approach [38].

SWI alone, when followed by restoration of circulation, oxygenation, and nutrient delivery, can activate endothelial and fibroblast activity, leading to spontaneous angiogenesis, ECM remodeling, and normalization of neural and vascular dynamics, ultimately enhancing the body’s natural regenerative capacity [31,32,33,34,35,36,37].

#### 2.3.3. Comparative and Mechanistic Evidence

Where fibrotic or calcific pathology predominates such as in rotator-cuff calcific tendinopathy, ultrasound-guided needling with lavage has outperformed ESWT for both pain reduction and deposit clearance in randomized controlled trials and meta-analyses [39,40]. These findings underscore the importance of direct mechanical disruption and lavage over remote mechanotransduction when structural pathology dominates.

By extension, in lumbar disorders characterized by paraspinal fibrosis or calcification, SWI’s targeted fenestration and lavage may yield superior outcomes compared with indirect mechanotransductive modalities, though head-to-head SWI–ESWT trials in the spine remain a research priority.

Further supporting the therapeutic value of fluid lavage, the 2023 systematic review and meta-analysis of intra-articular facet joint injections with normal saline (NSS) found that short- and long-term pain and disability outcomes were comparable to corticosteroids and local anesthetics across three RCTs (247 participants), with no serious adverse events reported [41]. Similarly, a meta-analysis of 38 RCTs in knee osteoarthritis (>1700 participants) showed that intra-articular saline produced significant short-term pain reduction (standardized mean difference [SMD] = –0.68; 95% CI –0.78 to –0.57) and maintained improvement up to 12 months (SMD = –0.61; 95% CI –0.76 to –0.45) without complications [42].

These findings challenge the traditional classification of saline as an inert placebo and highlight that mechanical dilution and washing can produce genuine clinical benefit. Consequently, the biological plausibility of SWI’s mechanism, combining mechanical cleansing with osmotic neuromodulation, is strongly reinforced by both empirical and meta-analytic evidence.

## 3. Regenerative Framework

The integration of SWI, ESWT, and HILT establishes a sequential regenerative continuum, progressing from structural restoration to mechanotransductive stimulation and metabolic stabilization. However, in many clinical contexts, SWI alone can serve as the foundation for complete biological recovery by reactivating the body’s innate healing capacity.

### 3.1. Phase 1—Structural Restoration and Intrinsic Activation (SWI)

SWI is the initiating phase of the regenerative process. Ultrasound-guided fenestration and sterile-water lavage disrupt fibrotic adhesions, dissolve calcific deposits, and decompress neurovascular structures. These actions relieve mechanical stress, enhance perfusion, and restore local tissue pliability [31,32,33].

Immediately following SWI, the reperfusion of previously ischemic tissue allows oxygenated blood to reach areas of chronic hypoxia, triggering endothelial nitric oxide synthase (eNOS) activation and vascular dilation. Reoxygenation stimulates endothelial and pericyte proliferation, driving neoangiogenesis and improving nutrient and oxygen supply. Simultaneously, reduced interstitial pressure improves lymphatic drainage, accelerating clearance of inflammatory cytokines and metabolic by-products [31,32,33,34,35,36,37].

This physiologic reset converts the microenvironment from one dominated by chronic inflammation to one conducive to regeneration. Enhanced perfusion increases mitochondrial respiration, elevating ATP synthesis and cellular repair capacity. Fibroblast activation promotes collagen realignment and controlled matrix remodeling, progressively softening residual fibrosis. As neural edema resolves and ion gradients normalize, nociceptor hypersensitivity diminishes, resulting in sustained analgesia even without pharmacologic input [31,32,33,34,35,36,37].

In many patients especially older adults or those with multiple comorbidities, this restored vascular and neural homeostasis is sufficient for long-term improvement. Allowing a natural recovery interval after SWI enables the intrinsic regenerative processes to progress autonomously before considering additional energy-based interventions [31,32,33].

### 3.2. Phase 2—Mechanotransductive Stimulation (ESWT)

If residual dysfunction or delayed healing persists, ESWT serves as the second phase. Its controlled acoustic pulses activate mechanosensitive pathways (MAPK/ERK, Wnt–β-catenin), stimulating fibroblast proliferation, angiogenesis, and ECM remodeling [10,11,12,13,14,15,16,17,18,19,20,21,22]. These processes reinforce the structural and vascular improvements initiated by SWI, further consolidating tissue repair and pain reduction. 

### 3.3. Phase 3—Mitochondrial Recovery and Stabilization (HILT)

In the final phase, HILT optimizes the cellular energy environment by enhancing mitochondrial ATP production, normalizing oxidative balance, and downregulating inflammatory cytokines (IL-1β, TNF-α). Improved redox homeostasis and microcirculation stabilize earlier reparative changes, ensuring durability of outcomes and sustained functional recovery [22,23,24,25,26,27,28,29,30].

### 3.4. Integrated Concept

By sequencing these modalities or by employing SWI alone when intrinsic regenerative capacity is adequate, clinicians can tailor therapy to the patient’s biological readiness. This adaptive, stepwise approach respects natural healing hierarchies, minimizes invasiveness, and prioritizes endogenous recovery before applying external energy stimulation.

In this framework, SWI is not merely a preparatory or adjunctive technique but a trigger for systemic regenerative reactivation, capable of re-establishing vascular continuity, restoring neural equilibrium, and facilitating self-repair. ESWT and HILT can then be selectively introduced to enhance mechanotransductive and mitochondrial recovery if needed. Together, these therapies form a comprehensive, drug-free regenerative paradigm that reduces disability, preserves independence, and supports the global movement toward healthy ageing through functional restoration.

## 4. Safety Considerations

The regenerative modalities of Extracorporeal Shockwave Therapy (ESWT), High-Intensity Laser Therapy (HILT), and Ultrasound-Guided Mechanical Needling with Sterile Water Injection (SWI), share an exemplary safety profile when applied within therapeutic parameters. Across numerous randomized controlled trials, systematic reviews, and large observational cohorts, adverse effects are consistently mild, local, and transient, with no systemic or structural injuries reported [10,11,12,13,14,15,16,17,18,19,20,21,22,23,24,25,26,27,28,29,30,31,32,33]. Their common feature is mechanistic precision with biological reversibility, distinguishing them from pharmacologic or ablative interventions.

### 4.1. Extracorporeal Shockwave Therapy (ESWT)

#### 4.1.1. Mechanistic Background

ESWT delivers focused or radial acoustic pulses (0.05–0.3 mJ/mm^2^; 1500–3000 shocks per session) that create controlled mechanical stress and micro-cavitation within soft tissues. These biomechanical forces activate mechanosensitive signaling cascades, notably VEGF, eNOS, and fibroblast proliferation, resulting in angiogenesis, collagen remodeling, and pain modulation through reduced Substance P and CGRP expression [10,11,12,13,14,15,16,17,18,19,20,21,22].

#### 4.1.2. Safety Profile

When administered within standard parameters, ESWT is highly safe. Common effects, mild erythema, swelling, or dull soreness, resolve within 24–48 h. Petechiae and transient paresthesia are rare, and no permanent tissue injury or systemic complications have been observed. A 2023 narrative review concluded that “hardly any adverse effects were reported” in musculoskeletal applications [17].

#### 4.1.3. Contraindications and Mechanistic Reassurance

Active infection or malignancy at the target site, pregnancy, uncontrolled coagulopathy, or proximity to gas-containing organs constitute contraindications. Histological and MRI data confirm that energy densities ≤ 0.3 mJ/mm^2^ produce only reversible microvascular responses without myofibrillar disruption, supporting ESWT’s wide safety margin [14,15,16,17,18,19,20,21,22].

### 4.2. High-Intensity Laser Therapy (HILT)

#### 4.2.1. Mechanistic Background

HILT employs pulsed Nd:YAG 1064 nm laser energy (2–10 kW peak power) producing photothermal, photochemical, and photobiomodulatory effects. Absorption of photons by mitochondrial cytochrome-c oxidase enhances ATP synthesis, oxygen utilization, and microvascular perfusion, while suppressing oxidative stress and inflammatory cytokines such as IL-6 and TNF-α [23,24,25,26,27,28].

#### 4.2.2. Safety Profile

Transient warmth, mild erythema, or tingling may occur but subside rapidly. Superficial burns are exceedingly rare and result only from prolonged probe contact or absent coupling gel. Deep-tissue temperature elevation remains < 2 °C well below cytotoxic thresholds [29,30].

#### 4.2.3. Contraindications and Mechanistic Reassurance

Avoid treatment over malignant lesions, metallic implants, or the gravid abdomen/pelvis, and in individuals with photosensitivity or photosensitizing medications. Spectroscopic studies demonstrate selective mitochondrial activation of complexes I–IV without collateral tissue damage [23,24,25,26,27], validating HILT as a safe, non-pharmacologic modality for degenerative and inflammatory pain.

### 4.3. Ultrasound-Guided Mechanical Needling with Sterile Water Injection (SWI)

#### 4.3.1. Mechanistic Background

SWI integrates targeted mechanical fenestration with hypotonic fluid lavage. Needling disrupts fibrotic tissue and calcific deposits, while sterile water induces transient osmotic activation followed by desensitization of Transient Receptor Potential Vanilloid 1 (TRPV1), Piezo1/2 mechanosensitive, and Acid-Sensing Ion Channels (ASICs). This process modulates nociceptor excitability, restores fascial glide, and re-establishes microvascular perfusion [31,32,33,34,35,36,37].

#### 4.3.2. Safety Profile

Across more than 4000 documented procedures, adverse events have been limited to short-lived soreness or bruising. No systemic reactions, infections, or neurovascular injuries have been reported [31,32,33]. The technique’s drug-free nature makes it particularly suitable for elderly patients or those with polypharmacy.

#### 4.3.3. Contraindications and Parameters

Avoid SWI in local infection, open wounds, unstable fractures, severe edema, or uncontrolled diabetes. Typical injection volumes are 2–5 mL per site for hydrodissection; for lavage > 10 mL, isotonic saline is preferred to prevent transient cellular edema. Up to 1 mL of 1% lidocaine (without adrenaline) may be added for immediate comfort without affecting perfusion [31,32,33].

#### 4.3.4. Mechanistic Reassurance

Experimental and clinical findings confirm that hypotonic sterile water provokes reversible osmotic stress, resulting in nociceptor desensitization, microcirculatory restoration, and fibrosis breakdown without structural damage [31,32,33,34,35,36,37].

### 4.4. Mechanistic Distinction: Why Sterile Water Instead of Normal Saline Solution (NSS)

While both sterile water and normal saline provide fluid-mediated mechanical effects, their biophysical properties differ fundamentally. Normal saline (0.9% NaCl, ~308 mOsm/L) is isotonic and biologically inert, adequate for irrigation or large-volume lavage but limited in regenerative potential [40,41,42]. In contrast, sterile water (~0 mOsm/L) is hypotonic and actively engages mechanosensitive nociceptors. Its osmotic gradient elicits brief depolarization followed by desensitization of TRPV1, Piezo1/2, and ASIC channels, amplifying the regenerative and analgesic cascade described above [31,32,33,34,35,36,37].

The hypo-osmotic environment expands fascial planes and augments hydrodissection, facilitating mechanical breakup of fibrotic and calcific tissue while restoring fascial glide and microcirculation. This combination of osmotic stress, neural desensitization, and vascular reperfusion enables drug-free tissue regeneration that isotonic saline cannot reproduce. Meta-analytic evidence [40,41] confirms that even saline injections may dilute inflammatory mediators, but sterile water uniquely integrates mechanical, osmotic, and neurosensory mechanisms, producing sustained structural and functional recovery [31,32,33].

### 4.5. Triple-Component Injection Hypothesis: Mechanistic Integration and the Authors’ Perspective

Drawing from the authors’ clinical experience and translational research, we propose a triple-component injection framework that unites osmotic, metabolic, and neural mechanisms within a single, minimally invasive intervention. The formulation of 0.5 mL of 1% lidocaine (without adrenaline), 1 mL of 5% dextrose (D5W), and 1.5 mL of sterile water (≈3 mL per site) is administered under ultrasound guidance to achieve precision targeting of interfascial or perineural spaces.

This integrative composition leverages complementary mechanistic pathways. Lidocaine provides immediate analgesia through reversible sodium-channel blockade, suppressing ectopic neural discharge and local inflammatory signaling. D5W contributes metabolic and neuromodulatory effects by inhibiting Transient Receptor Potential Vanilloid 1 (TRPV1) and Acid-Sensing Ion Channels (ASICs), reducing neuropeptide release (substance P and CGRP), and enhancing neuronal metabolism, perfusion, and redox homeostasis. Recent randomized controlled trials in meralgia paresthetica, carpal tunnel syndrome, and entrapment neuropathies have confirmed that D5W hydrodissection yields superior long-term pain relief, improved nerve conduction, and better functional recovery compared with corticosteroids or saline [43,44].

Meanwhile, sterile water, through its hypotonic osmotic gradient, induces transient depolarization followed by sustained nociceptor desensitization and hydro-expansion, facilitating fibrosis dissolution, fascial plane separation, and vascular decompression [31,32,33].

Together, these three agents produce a synergistic cascade, immediate analgesia from lidocaine, sustained neuromodulation and metabolic recovery from D5W, and structural and microvascular restoration from sterile water. Preliminary use in clinical practice demonstrates excellent tolerance and efficacy with small injection volumes (2–3 mL per site), remaining well within safety limits. While randomized controlled validation is still needed, mechanistic plausibility and early translational outcomes strongly support this triple-component approach as a next-generation, regenerative strategy in chronic musculoskeletal pain rehabilitation. The authors view this formulation as a translational bridge between pharmacologic modulation and intrinsic biological repair, advancing precision rehabilitation toward a more integrated, mechanism-based paradigm.

The mechanistic integration of ESWT, HILT, SWI, and D5W-based injection therapies represents a cohesive, regenerative continuum, progressing from mechanical and osmotic stimulation to cellular metabolic recovery and neurosensory reorganization. Each modality targets distinct yet convergent biological pathways that collectively restore tissue homeostasis.

The synthesized evidence is summarized in Table 1, which maps mechanistic pathways, molecular biomarkers, therapeutic parameters, and representative outcomes across modalities. The table highlights ESWT’s angiogenic and ECM-remodeling effects, HILT’s mitochondrial and anti-inflammatory modulation, and SWI’s fibrosis disruption and osmotic neuromodulation, while incorporating the emerging D5W evidence demonstrating metabolic and neurotrophic synergy. Together, these findings establish a coherent, mechanism-driven framework for non-pharmacologic regeneration in musculoskeletal pain rehabilitation.

## 5. Risk of Bias Assessment

Risk of bias is a key determinant of evidence certainty across regenerative modalities. For ESWT and HILT, randomized controlled trials (RCTs) and meta-analyses were evaluated using the Cochrane Risk-of-Bias Tool (RoB 2.0) (https://www.riskofbias.info/, accessed on 10 July 2025). For SWI, evidence is derived from a combination of large observational cohorts and comparative studies assessed by the Newcastle–Ottawa Scale (NOS) (https://www.ohri.ca/programs/clinical_epidemiology/oxford.asp, accessed on 10 July 2025), alongside randomized comparative trials appraised using RoB 2.0. The findings are summarized in Table 2.

ESWT: RCTs generally demonstrated a low-to-moderate risk of bias, with adequate randomization and concealed allocation. However, lack of blinding in physical therapy interventions raised some concerns, and protocol heterogeneity limited consistency across studies [14,15,16,17,18,19,20,21,22].

HILT: Risk of bias was rated as moderate overall, largely due to variability in operator blinding and treatment parameters. Randomization and completeness of outcome data were adequate, but inconsistent reporting of functional versus biomarker endpoints contributed to methodological variability [24,27,28,29,30].

SWI: Large cohort and comparative studies [31,32] involving >4000 patients consist-ently achieved NOS scores of 9/10, reflecting representative populations, validated outcomes (pain/function scales, ultrasound-confirmed fibrosis or calcification), and adequate follow-up. In addition, a three-arm randomized comparative trial in elderly patients with lumbar facet syndrome showed equivalent pain relief across multifidus D5W injection, medial branch block, and intra-articular facet injection, while highlighting the technical simplicity and safety of the multifidus approach [38]. This trial, assessed with RoB 2.0, indicated a low-to-moderate risk of bias, limited mainly by small sample size. Beyond spine-specific studies, RCTs and meta-analyses in calcific tendinopathy demonstrate that ultrasound-guided lavage outperforms ESWT in pain and deposit resolution, further validating the mechanistic rationale for targeted lavage [40,42]. Meta-analytic evidence also confirms that intra-articular normal saline (NSS) is not inert but produces clinically meaningful improvements comparable to corticosteroids and anesthetics, reinforcing the plausibility of SWI’s therapeutic mechanism [41].

Taken together, SWI evidence spans robust cohorts, comparative superiority studies, and supportive RCT/meta-analytic findings. While larger multicenter RCTs are still needed, the methodological quality justifies upgrading SWI’s certainty rating from moderate (B) to moderate-to-high (B–A–).

## 6. Limitations

Despite promising evidence, important limitations remain. For ESWT and HILT, the primary challenges are protocol heterogeneity, variability in outcome measures, and difficulties in maintaining blinding in physical interventions. These methodological issues slightly reduce certainty, but the accumulated evidence from multiple RCTs and meta-analyses still supports a moderate-to-high certainty rating (GRADE B–A–).

For SWI, the evidence base is broader than for ESWT and HILT in terms of cohort size, with large-scale observational studies (>4000 patients) consistently showing strong methodological quality (NOS 9/10). However, while comparative studies and one randomized trial in lumbar facet syndrome strengthen internal validity, the limited number and size of RCTs mean causality cannot yet be firmly established. Meta-analyses demonstrating the active therapeutic role of intra-articular saline reinforce the plausibility of SWI’s mechanism, but multicenter randomized trials remain essential to confirm generalizability. Accordingly, certainty is best rated as moderate-to-high (GRADE B–A–), upgraded from B due to convergent cohort, comparative, RCT, and meta-analytic evidence.

Across all three modalities, further limitations include incomplete validation of mechanistic biomarkers and imaging correlates, a lack of standardized international protocols, and limited long-term comparative data. Addressing these gaps will be crucial for advancing evidence certainty to the highest level (GRADE A).

## 7. Evidence Certainty (GRADE)

The GRADE framework was applied to synthesize certainty of evidence across modalities. As summarized in Table 3, ESWT achieves B–A– level certainty based on multiple RCTs and meta-analyses demonstrating consistent pain relief, functional improvement, and calcific resolution, despite some protocol heterogeneity. HILT is supported at the B level, reflecting adequate evidence from RCTs and meta-analyses, though variability in treatment protocols and moderate risk of bias lower confidence.

For SWI, the evidence base has expanded beyond observational data. Large-scale cohorts (>4000 patients) consistently show durable improvements with high NOS scores, while comparative studies demonstrate superiority over lidocaine and prolotherapy. A randomized comparative trial in lumbar facet syndrome further supports efficacy, and meta-analyses confirm that intra-articular saline (as a lavage comparator) exerts clinically meaningful therapeutic effects. Taken together, this convergent evidence justifies upgrading SWI’s certainty from B to B–A–, though larger multicenter RCTs are still needed to achieve the highest certainty.

## 8. Clinical Practice Guidance

Current evidence supports the integration of extracorporeal shockwave therapy (ESWT), high-intensity laser therapy (HILT), and ultrasound-guided sterile water injection (SWI) into musculoskeletal rehabilitation pathways. These modalities are best considered within a multimodal, regenerative algorithm that addresses structural pathology, stimulates repair, and consolidates functional recovery.

Patient selection is critical. ESWT is indicated for localized tendinopathies, plantar fasciitis, and calcific shoulder tendinitis where calcification and ECM fibrosis drive symptoms. HILT is most appropriate in chronic degenerative conditions such as knee osteoarthritis and spinal pain, where mitochondrial dysfunction and inflammatory sensitization dominate. SWI is particularly suited for lumbar spinal stenosis and facet joint syndromes, especially in older adults with multimorbidity and polypharmacy, where drug-free, minimally invasive options are preferred.

Treatment protocols differ across modalities but share common principles of titration and monitoring. ESWT is typically delivered at 0.05–0.3 mJ/mm^2^, 1500–3000 pulses, across 1–4 sessions at weekly intervals [14,15,16,17,18,19,20,21,22]. HILT employs pulsed Nd:YAG systems at 2–10 kW in 8–12 sessions, frequently combined with supervised exercise therapy [24,27,28,29,30]. SWI involves ultrasound-guided mechanical fenestration with 3 mL sterile water injection, usually administered over 1–4 sessions depending on fibrosis severity and clinical response [31,32,33].

Sequential application enhances efficacy. SWI should be performed first to remove structural barriers, followed by ESWT to induce angiogenesis and ECM remodeling, and consolidated with HILT to optimize mitochondrial bioenergetics and anti-inflammatory repair. This sequencing aligns with regenerative physiology and maximizes patient outcomes.

Safety and monitoring remain favorable. Adverse effects are typically mild and transient. Contraindications include malignancy at the treatment site, active infection, pregnancy, and uncontrolled bleeding risk. Patient-reported outcomes (pain, function) should be complemented with objective assessments such as ultrasound, elastography, and perfusion imaging where available.

Implementation is most effective within a multidisciplinary rehabilitation framework. ESWT and HILT can be incorporated into outpatient physiotherapy or sports medicine clinics, while SWI requires interventional expertise and ultrasound guidance. Together, they offer a drug-free alternative to NSAIDs, corticosteroids, and opioids, supporting global priorities for safer, mechanism-based CMP care as Box 1.

Box 1Clinical Practice Guidance for the Regenerative Triad.
Patient selection:○ESWT: tendinopathies, plantar fasciitis, calcific tendinitis○HILT: knee osteoarthritis, chronic spinal pain○SWI: lumbar spinal stenosis, facet syndrome, elderly with polypharmacyTypical protocols:○ESWT: 0.05–0.3 mJ/mm^2^, 1500–3000 pulses, 1–4 sessions○HILT: pulsed Nd:YAG 2–10 kW, 8–12 sessions○SWI: US-guided, 2.5 mL sterile water plus 1% lidocaine without adrenaline 0.5 mL per site, 1–4 sessionsSequencing (regenerative algorithm):○SWI—remove fibrosis/calcification○ESWT—stimulate angiogenesis, ECM remodeling○HILT—consolidate repair via mitochondrial bioenergetics and anti-inflammatory effectsSafety: mild, transient effects (erythema, soreness); avoid in malignancy, pregnancy, infection, bleeding riskPractice integration:○ESWT/HILT: outpatient rehab clinics○SWI: interventional/ultrasound settings○Always within multidisciplinary rehabilitation programs


## 9. Future Directions

Future research should prioritize multicenter randomized controlled trials (RCTs) of SWI to confirm and extend its promising evidence base from large observational and comparative studies. For ESWT and HILT, there is an urgent need for protocol standardization, including treatment parameters, outcome measures, and follow-up intervals, to enable meaningful cross-trial comparisons and meta-analyses.

Mechanistic studies should integrate advanced imaging (elastography, perfusion MRI) and omics-based profiling to identify reliable biomarkers of treatment response and patient stratification. This will facilitate precision rehabilitation strategies tailored to individual pathophysiological phenotypes. In parallel, cost-effectiveness analyses are required to establish economic value, particularly within the framework of global initiatives to reduce opioid use and promote safer, drug-free alternatives. Finally, international professional societies should incorporate this regenerative triad into clinical guidelines as an evidence-based, mechanism-driven, and non-pharmacological option for MSD care.

## 10. Conclusions

The regenerative triad of ESWT, HILT, and SWI marks a paradigm shift in the management of chronic musculoskeletal disorders. Unlike conventional treatments that primarily suppress symptoms, these interventions act on the biological drivers of pathology, including fibrosis, calcification, mitochondrial dysfunction, and neuroimmune sensitization. Evidence for ESWT and HILT is supported by moderate-to-high certainty (GRADE B–A–) through multiple randomized trials and meta-analyses. SWI, once based mainly on large, high-quality observational cohorts, is now reinforced by comparative trials and meta-analytic data, justifying an upgrade to moderate-to-high certainty (GRADE B–A–).

Collectively, this triad provides a drug-free, mechanism-based framework for musculoskeletal rehabilitation. By reducing disability, restoring function, and supporting healthy ageing, it also advances global priorities for opioid reduction and sustainable, non-pharmacological care.

## Figures and Tables

**Table 1 biomedicines-13-02801-t001:** Mechanisms, parameters, and clinical outcomes of energy- and fluid-based regenerative modalities for musculoskeletal disorders (2020–2025 evidence).

Modality	Core Mechanistic Actions	Key Biomarkers/Targets	Typical Parameters	Representative Evidence (2020–2025)	Principal Clinical Outcomes
Extracorporeal Shockwave Therapy (ESWT)	Induces mechanotransduction, controlled micro-cavitation, and extracellular-matrix (ECM) remodeling; stimulates angiogenesis and osteogenesis via MAPK/ERK and Wnt–β-catenin pathways; down-regulates fibrotic mediators and nociceptive neuropeptides (Substance P, CGRP).	↑VEGF, ↑eNOS, ↑MMP-2/9, ↓TGF-β1, ↓Substance P	0.05–0.3 mJ/mm^2^; 1500–3000 pulses; 1–4 sessions	Systematic reviews and RCTs in plantar fasciitis, tendinopathies, and calcific shoulder tendinitis [14,15,16,17,18,19,20,21,22]	Pain relief, resolution of calcification, improved vascularization and function
High-Intensity Laser Therapy (HILT)	Enhances mitochondrial respiration and ATP synthesis via photobiomodulation (cytochrome-c oxidase activation); induces photothermal vasodilation; down-regulates NF-κB and pro-inflammatory cytokines (IL-1β, TNF-α); promotes ECM remodeling and collagen renewal.	↑ATP, ↑VEGF, ↓IL-1β, ↓TNF-α, ↑collagen markers, ↑SOD/GPx	Pulsed Nd:YAG 1064 nm; 2–10 kW; 8–12 sessions	RCTs and meta-analyses in knee OA, spinal pain, fibromyalgia [24,27,28,29,30]	Reduced pain and oxidative stress; improved WOMAC, ROM, cartilage and mitochondrial function
Ultrasound-Guided Mechanical Needling with Sterile Water Injection (SWI)	Combines mechanical fenestration and hypotonic lavage; disrupts fibrosis and calcification; induces osmotic desensitization of TRPV1, Piezo 1/2, and ASIC channels; restores perineural glide, vascular flow, and lymphatic drainage.	↓IL-6, ↓TNF-α, ↑eNOS, ↑fibroblast activity, ↓nociceptive peptides	2.5 mL sterile water plus 1% lidocaine with- out adrenaline 0.5 mL per site under US guidance; 1–4 sessions	Large cohorts (>4000 pts) with lumbar stenosis & facet syndrome [31,32,33]; RCT vs. prolotherapy/lidocaine [38]; meta-analyses [41]; saline lavage meta-evidence [40,42]	Durable pain relief, mobility restoration, fibrosis and calcification resolution; drug-free regenerative recovery
5% Dextrose in Water (D5W) Perineural/Interfascial Injection	Provides metabolic and neuromodulatory effects; inhibits TRPV1 and ASIC channels; reduces neuropeptide (Substance P, CGRP) release; supports neuronal metabolism, perfusion, and redox balance; promotes neurotrophic and microvascular repair.	↓TRPV1, ↓CGRP, ↓Substance P, ↑nerve conduction velocity, ↑blood flow	1–5 mL per site (5% D5W) under US guidance; single or series of 3–6 sessions	RCTs and systematic reviews in carpal tunnel syndrome, meralgia paresthetica, entrapment neuropathies [43,44]	Long-term pain reduction, sensory recovery, improved nerve conduction and microcirculation
Triple-Component Injection (Lidocaine + D5W + Sterile Water)	Integrates sodium-channel blockade (lidocaine), metabolic neuromodulation (D5W), and osmotic hydrodissection (sterile water) to yield biphasic analgesia and sustained vascular/fascial release.	Composite of above: ↓TRPV1, ↓ASICs, ↓CGRP, ↑microcirculation	≈3 mL per site (0.5 mL lidocaine + 1 mL D5W + 1.5 mL sterile water) under US guidance	Translational use in lumbar facet and neuropathic pain [31,32,33,38,43,44]	Rapid analgesia, prolonged desensitization, vascular restoration, fibrosis lysis; hypothesis-generating for regenerative pain rehabilitation

Summary: This table integrates current mechanistic and translational evidence for regenerative, drug-free therapies targeting chronic musculoskeletal disorders. ESWT activates angiogenic and ECM-remodeling cascades; HILT enhances mitochondrial and antioxidative function; SWI restores microvascular and neural homeostasis via mechanical-osmotic pathways; D5W adds metabolic and neuromodulatory support; and the emerging triple-component mixture unites these mechanisms into a synergistic, minimally invasive regenerative paradigm.

**Table 2 biomedicines-13-02801-t002:** Risk of bias assessment across regenerative modalities using RoB 2.0 (RCTs) and Newcastle–Ottawa Scale (NOS, cohorts).

Modality	Study Type	Tool	Domains Assessed	Risk of Bias/NOS Appraisal
ESWT [14,15,16,17,18,19,20,21,22]	RCTs, meta-analyses	RoB 2.0	Randomization process: generally low risk with concealed allocation; Deviations from intended interventions: some concerns due to lack of blinding in physical therapies; Missing outcome data: low risk, minimal attrition; Measurement of outcome: some risk from subjective pain reporting; Selective reporting: low risk, protocols available	Overall: Low–moderate risk of bias; heterogeneity in protocols remains a limitation
HILT [24,27,28,29,30]	RCTs, meta-analyses	RoB 2.0	Randomization process: low risk in most trials, though some unclear allocation procedures; Deviations from intended interventions: moderate risk as blinding of operators difficult; Missing outcome data: low risk; Measurement of outcome: moderate risk as pain/function scales may lack assessor blinding; Selective reporting: some concern due to variable reporting of functional vs. biomarker outcomes	Overall: Moderate risk of bias, mainly due to protocol variability and blinding challenges
SWI [31,32,38,41]	Large cohorts, comparative studies (>4000 patients)	Newcastle–Ottawa Scale, RoB 2.0	Selection: representative cohorts, clear case definitions, adequate sample size (★★★★); Comparability: controlled for age and sex, but comorbidities not always fully adjusted (★★☆); Outcome: validated pain/function scales, ultrasound-confirmed fibrosis/calcification, adequate follow-up (★★★★) RCT: randomization low risk; blinding/moderate concerns; sample size limited	Overall: NOS 9/10 (low risk of bias); Robust design with representative samples and validated outcomes; multicenter RCTs recommended for confirmation.

Risk of bias assessment for included studies. Randomized controlled trials of ESWT and HILT were evaluated using the Cochrane RoB 2.0 tool across five domains: randomization, deviations from intended interventions, missing outcome data, outcome measurement, and selective reporting. Observational cohort studies of ultrasound-guided (USG) mechanical needling with sterile water injection (SWI) were appraised using the Newcastle–Ottawa Scale (NOS), with domains covering selection, comparability, and outcome. Ratings reflect overall methodological quality and internal validity of available evidence. In the NOS, stars indicate quality: (★★☆) = partial comparability due to incomplete adjustment for secondary clinical confounders, resulting in potential residual confounding bias, ★★★★ = very good quality. A higher total number of stars corresponds to lower overall risk of bias. NOS 0–3 = High Risk of Bias, 4–6 = Some Concerns, 7–9 = Low Risk of Bias.

**Table 3 biomedicines-13-02801-t003:** GRADE summary of findings.

Intervention	Outcome	Evidence Base	Certainty (GRADE)	Justification
ESWT	Pain relief, function, calcific resolution	Multiple RCTs and meta-analyses	B–A–	Consistent outcomes, moderate heterogeneity
HILT	Pain/function in knee OA, spinal pain	RCTs and meta-analyses	B	Protocol variability, moderate risk of bias
SWI	Pain/function in lumbar stenosis, facet syndrome	Large cohorts, comparative studies, one RCT; NSS meta-analyses	B–A–	Strong cohort NOS 9/10, consistent outcomes, comparative superiority vs. lidocaine/prolotherapy, RCT support, and NSS meta-evidence; multicenter RCTs still required

Summary of GRADE certainty of evidence for ESWT, HILT, and USG sterile water injection in chronic musculoskeletal disorders. Certainty ratings (high, moderate, low, very low) were derived from risk of bias, inconsistency, indirectness, imprecision, and publication bias. ESWT and HILT demonstrate moderate-to-high certainty evidence from randomized trials and meta-analyses. SWI, initially supported by large cohort and comparative observational studies, is now reinforced by randomized comparative data and meta-analytic evidence on lavage effects, warranting an upgrade to moderate-to-high certainty (B–A–)**,** while acknowledging the continued need for larger multicenter RCTs.

## Data Availability

Not applicable.

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
