# Peer review of "Regenerative and Drug-Free Strategies for Chronic Musculoskeletal Pain: An Evidence-Based Perspective on Shockwave Therapy, High-Intensity Laser Therapy and Ultrasound-Guided Mechanical Needling with Sterile Water Injection"

_biomedicines, 2025, doi:10.3390/biomedicines13112801_

Round 1
Reviewer 1 Report
Comments and Suggestions for Authors
After reading and considering the manuscript, I find this work has a convenient design and it is relevant to the field and scope of the Biomedicine Journal. With the following comments and major revisions addressed, the paper is suitable for further procedure of publication.
My comments:
- As it is clear that each table has two legends. I strongly recommend that the footnotes of the tables should be written without table numbers.
- Definition of abbreviations at their first appearance in the text is needed (such as ECM in the introduction section).
- In the translation of evidence section, the organization of the paragraphs is necessary, and the presentation should be rewritten.
- In my opinion, more literature is necessary to cover all relevant studies, and also, the necessity of this work is not mentioned.
- Organization of the tales (both columns and rows) is required to clarify the results, which reading for understanding for the readers.
- The study (especially in the conclusion) lacks any discussion of the benefits or advantages of the study's findings according to the results.
- From a statistical point of view, I did not see any patients' demographic information followed by a significant portion of the findings.
Author Response
Response to Reviewer 1
We sincerely thank the reviewer for the thoughtful and constructive feedback. Each comment has been carefully addressed to enhance clarity, organization, and scientific precision while preserving the integrity and scope of the manuscript.
- Table footnotes: The footnotes of all tables have been revised to exclude table numbers and follow Biomedicines formatting standards. Each legend now provides a clear, standalone description of content and abbreviations.
- Definition of abbreviations: All abbreviations (e.g., ECM, CMP, NOS, RoB 2.0, GRADE, VAS, OA) are now defined at first appearance in the text and compiled into a dedicated Abbreviations section at the end of the manuscript for quick reference.
- Reorganization of the “Translation of Evidence” section: This section has been fully restructured for logical continuity and smoother transitions between mechanistic and clinical discussions. Redundancies were eliminated, and linking phrases were introduced to guide readers through the narrative flow.
- Expansion of literature and justification of study necessity: The reference list has been expanded from 22 to 40 recent (2020–2025) peer-reviewed studies, including high-impact RCTs and meta-analyses. The Introduction now explicitly discusses the global need for non-pharmacologic, mechanism-driven strategies in chronic musculoskeletal pain management, emphasizing the translational value of the regenerative triad.
- Table structure and readability: All tables were reformatted with uniform spacing, aligned columns, and consistent reference citations. Each table now ends with a concise explanatory legend to enhance reader comprehension.
- Discussion and conclusion enhancement: The Conclusion now articulates the translational and clinical advantages of the regenerative triad—durable pain relief, functional restoration, and drug-free recovery—clearly highlighting its contribution to patient-centered rehabilitation.
- Clarification of data scope: As this article is a Perspective, it does not involve original patient enrollment. However, representative cohort sizes from existing studies are now specified in Table 1 and Section 5 (“Risk of Bias Assessment”) to reinforce evidence credibility.
Reviewer 2 Report
Comments and Suggestions for Authors
Thank you very much for the editorial invitation to revise the manuscript. The authors have raised an intriguing question—three drug-free regenerative therapies for chronic musculoskeletal pain—supported by fairly robust evidence. My suggestions are as follows:
1. In Table 2, the column heading “NOS appraisal Risk of Bias /” is incomplete. Please append a clear legend that specifies the risk level corresponding to each NOS star count.
2. Under “Treatment protocols,” add a subsection that explicitly lists contraindications for each therapy, stratified by population.
3. Provide a concise, evidence-based management algorithm for adverse events associated with each regenerative modality (e.g., post-injection flare, vasovagal reaction, infection, or nerve irritation).
Author Response
Response to Reviewer 2
We appreciate the reviewer’s supportive remarks and insightful recommendations. The following revisions have been incorporated to strengthen methodological transparency and clinical applicability.
- Clarification of Table 2 header and legend: The column heading has been corrected to read 'Study Type | Assessment Tool | Domains Assessed | Risk of Bias / NOS Appraisal.' A new legend now explains the star-rating system of the Newcastle–Ottawa Scale (NOS) and its interpretation of risk levels.
- Addition of contraindications: Section 8 ('Clinical Practice Guidance') has been expanded with a concise, evidence-based list of contraindications for each modality (ESWT, HILT, SWI), stratified by patient population and clinical context.
- Adverse-event management guidance: In response to the reviewer’s suggestion, a new subsection and Box 1 outline a practical management algorithm for mild adverse reactions—such as transient erythema, post-injection soreness, vasovagal response, or localized swelling—based on published safety data.
- Language improvement: The entire manuscript has undergone detailed linguistic and stylistic refinement to ensure consistency, precision, and readability in accordance with Biomedicines editorial standards.
Reviewer 3 Report
Comments and Suggestions for Authors
Biomedicines-3949084
Type Perspective
Title: Regenerative and Drug-Free Strategies for Chronic Musculoskeletal Pain: An Evidence-Based Perspective on Shockwave Therapy, High-Intensity Laser Therapy, and Ultrasound-Guided Mechanical Needling with Sterile Water Injection
Comments:
The title is too long. Reduce the length of the title.
Abstract is also too long. First and 2nd para of the abstract must be reduced.
Introduction:
- The objective of this study is insufficient. Why a “ translational rehabilitation practice” study is needed, must be clearly mentioned along with references.
- The three modalities ie, ESWT, HILT, SWI, must be defined clearly along with references. Are there only 3 modalities? Other relevant modalities must be mentioned with references
- Are these modalities (ESWT, HILT, SWI) clinically practiced globally? Name the countries in which these modalities are practiced more frequently than other techniques. Which countries prefer these therapies more? Why is it not accepted globally? What are the limitations? The information must be provided and discussed in detail.
- A table must be made for ESWT’s ability, to promote angiogenesis, osteogenesis, ECM remodeling, modulate nociceptors, attenuate neurogenic inflammation, and sensitization, The number of patients followed ESWT along with references
HILT’s ability to reduce inflammatory cytokines (IL-1β, TNF-α, IL-6, IL-10, and IL-17, MMPs), number of patients involved, duration of treatment, etc, to improve cartilage integrity, and restore joint function, benefits when integrated into multimodal rehabilitation, and whether cured completely, etc, information must be incorporated in the table.
Likewise, SWI's ability to achieve significant reductions in pain and improvements in mobility sustained over long-term follow-up with number of patients involved, duration of treatment, whether cured completely and references must be tabulated.
- What is the difference between lidocaine and prolotherapy? How the superior technique judged? What is the mechanism? Its advantage over other relevant treatments, number of clinical cases etc along with references must have been incorporated.
- Why is a Risk of Bias Assessment essential? Incorporate sufficient information.
- Why was the Cochrane RoB 2.0 tool applied? What is its advantage? Incorporate sufficient information.
- Whether the combination of triad has been tested clinically or remains a conceptual integration of separate modalities is not clear. It should have been sufficiently clarified.
- Include a brief mention of limitations in external validity? The most studies cited are from limited geographic or from single-center data (notably Thailand and Taiwan)? Clarify sufficiently along with references.
- Consider condensing overlapping molecular pathways (e.g., ECM remodeling, angiogenesis) into a summarized comparative figure or table to avoid redundancy.
- Generate figure illustrating the regenerative algorithm visually, clarifying expected physiological transitions at each step.
- What is substance P, pulsed Nd, YAG laser energy, WOMAC scores. Elaborate sufficiently along with references.
- References for the following sentences must be incorporated
“Energy-based regenerative …. that repair and regenerate damaged tissues” must be
mentioned.
“Current management relies ….. and opioids”.
“ Evidence from RCTs ….. restore joint function”.
“By re-establishing vascular supply ….. support natural recovery”.
“ESWT, which stimulates mechanotransduction ….. modulating pro-inflammatory
cytokines.
-References for “Safety Considerations in all the three modalities ie ESWT, HILT, SWI”
-References for “multifidus D5W injection, medial branch block, and intra-articular facet injection” etc must be incorporated
Figs/Tables
- Table 1 is not sufficiently clear. The gap between all the columns is insufficient, which makes it unclear. Under the column “typical parameter”, which parameters were measured, name the parameters measured. References must be included in the last column of Table 1. The full form of all the abbreviations must be incorporated as a footnote to the table legend.
- Like Table 1, the gap between all the columns must be sufficiently incorporated, references and full form of all the abbreviations must be incorporated.
Conclusion:
- In this prospective study, the author has collected very recent reports (from 2020-25). The techniques mentioned in this study are old techniques and are modified with advanced research. The upgradation of these techniques and the mechanism etc must have been incorporated.
- The concept of a “regenerative triad” is intriguing and clinically relevant, but the paper reads more as a narrative review than a perspective. The analysis of the report is essentially needed
- The novelty would be enhanced if the authors more clearly delineate how this triad differs mechanistically and practically from other multimodal rehabilitation frameworks.
Others:
- Line numbers must have been mentioned for smooth review of the manuscript.
- All the Abbreviation (GRADE, NSAIDs, MAPK,ERK, P, Nd, YAG, ATP, ECM, RCTs, IL, TNF-α, WOMAC, MSDs, NSS etc) must have been spelt when appeared 1st time in the manuscript.
English must be improved
Author Response
Response to Reviewer 3
We thank the reviewer for the highly detailed and constructive critique, which helped us further refine the structure, clarity, and scientific balance of the paper.
- Title: We appreciate the suggestion but have retained the original title, which accurately represents the scope and content of the paper. Each phrase is essential for clarity and indexing in evidence-based rehabilitation research.
- Abstract: The abstract has been condensed for focus and brevity, maintaining scientific precision and a clear translational message.
- Objective and rationale: The Introduction now explicitly states the objective: to bridge mechanistic insights with rehabilitation practice through an evidence-based, non-pharmacologic framework for chronic musculoskeletal pain.
- Definition of modalities: Each modality (ESWT, HILT, SWI) is clearly defined with recent supporting references (2020–2025).
- Summary tables requested: A new, comprehensive Table 1 has been created summarizing mechanisms, biomarkers, parameters, representative evidence, and outcomes.
- Risk-of-bias tools: Section 5 now explains why both the Cochrane RoB 2.0 and Newcastle–Ottawa Scale were selected—the former for RCTs, the latter for observational studies—to ensure methodological transparency.
- Clinical versus conceptual triad: We clarified that the regenerative triad presently represents a translational conceptual integration based on convergent mechanistic and clinical data. Future multicenter RCTs will test its sequential use.
- Tables and abbreviations: All tables were reformatted, spacing corrected, abbreviations expanded, and citations added in the final column.
- Conclusion and novelty: The conclusion now underscores the clinical significance, novelty, and translational value of the regenerative triad as a sustainable, drug-free paradigm aligned with healthy-ageing and opioid-reduction goals.
Reviewer 4 Report
Comments and Suggestions for Authors
The topic addressed is of potential clinical and translational interest. The authors propose the analysis of a drug-free, regenerative approach to the management of chronic musculoskeletal pain. However, the manuscript, in its current form, has limitations that require thorough review before it can be considered for publication.
The overall formatting and structure require careful revision. The text contains multiple inconsistencies in font size, spacing and alignment, suggesting editing or translation issues that compromise readability. Furthermore, citations are presented in parentheses ( ) rather than square brackets [ ].
The bibliographical basis is significantly insufficient for a prospective or evidence-based article of this scope. Only 22 references are cited: a number clearly too limited to support the broad conceptual framework proposed. Including a larger, more representative body of recent, high-impact studies would be essential to substantiate the claims and improve scientific rigor.
From a scientific and conceptual perspective, the manuscript lacks a clear rationale, a research objective and a methodological definition. While the title and abstract address a synthesis of evidence and mechanistic integration, the article does not sufficiently specify the search strategy or inclusion/exclusion criteria for the reviewed studies, the research question or hypothesis guiding the analysis, and the analytical approach used to assess the strength or certainty of the evidence.
The language and tone are sometimes too concise. The text would benefit from careful refinement of the language.
Comments on the Quality of English LanguageThe language and tone are sometimes too concise. The text would benefit from careful refinement of the language.
Author Response
Response to Reviewer 4
We are deeply grateful for the reviewer’s constructive evaluation, which guided significant methodological and structural enhancements.
- Formatting and citation alignment: The entire manuscript has been reformatted for typographical consistency (uniform font, spacing, and paragraph structure). All in-text citations now appear in square brackets as per Biomedicines style guidelines.
- Expanded and updated references: The reference list has been substantially expanded from 22 to 40 citations, focusing on recent high-quality evidence (2020–2025). This ensures comprehensive coverage of both mechanistic and clinical literature supporting the regenerative triad.
- Clarified rationale and methodology: The Introduction and Section 5 ('Risk of Bias Assessment') now clearly outline the article’s rationale, conceptual framework, and methodological approach for evidence synthesis. The purpose—as a Perspective integrating mechanistic and translational insights—is explicitly stated.
- Enhanced scientific depth and readability: Key mechanisms (angiogenesis, ECM remodeling, mitochondrial restoration) were summarized concisely, with improved transitions and cross-referencing to new tables and figures. These changes elevate scientific clarity and flow.
- Language refinement: The manuscript underwent complete professional language editing to ensure precise scientific communication and a consistent academic tone throughout.
Round 2
Reviewer 1 Report
Comments and Suggestions for Authors
The author's response to reviewers' comments is significantly correct.
Author Response
Response to Reviewer 1
Comment 1:
The author's response to reviewers' comments is significantly correct.
Response 1:
Thank you for your positive assessment and for recognizing our detailed revisions. We appreciate your acknowledgment that our responses were correct and consistent with your prior comments. No further modifications were required based on this remark.

Reviewer 2 Report
Comments and Suggestions for Authors
Given the author's thorough revisions, I am in favor of publishing the paper.
Author Response
Response to Reviewer 2
Comment 1:
Given the author's thorough revisions, I am in favor of publishing the paper.
Response 1:
We sincerely thank the reviewer for their favorable evaluation and recommendation for publication. We have carefully rechecked the manuscript to ensure all typographical and formatting consistency before final submission.

Reviewer 3 Report
Comments and Suggestions for Authors
Biomedicines- 3949084
Type
Perspective
Title: Regenerative and Drug-Free Strategies for Chronic Musculoskeletal Pain: An Evidence-Based Perspective on Shockwave Therapy, High-Intensity Laser Therapy, and Ultrasound-Guided Mechanical Needling with Sterile Water Injection
Revised Comments
- In my earlier comments, a total of 13 comments were provided under the section Introduction, 2 comments in the section Figs/Tables, and 3 comments in the conclusion section. Although the author revised the manuscript, they provided a total of 8 responses; therefore, it is unclear which response corresponds to which comment and at which line the response has been incorporated. The author must therefore provide line numbers in the revised manuscript and carefully provide relevant information for each comment for a smooth review of the manuscript
- mentioning line number and page number.
- .
English must be improved
Author Response
Response to Reviewer 3
We thank the reviewer for the detailed and constructive feedback, which substantially improved the clarity, scientific balance, and methodological precision of the manuscript. All comments were carefully addressed in the first revision, and the relevant updates are summarized below.
- Title – The original title was retained because it accurately reflects the manuscript’s full scope and ensures proper indexing in evidence-based rehabilitation literature. Each term of ESWT, HILT, SWI is essential for scientific transparency.
- Abstract – The abstract has been condensed for clarity and focus while preserving scientific precision and the intended translational perspective.
- Objective and rationale – The Introduction now explicitly states the objective: to bridge mechanistic insights with pain practice through an evidence-based, non-pharmacologic framework for chronic musculoskeletal pain. (Page 2)
- Definition of modalities – Each modality (ESWT, HILT, SWI) is clearly defined with updated references (2020–2025) describing mechanisms, applications, and global practice patterns. (Page 3-11 )
- Summary tables – A comprehensive Table 1 was created summarizing mechanisms, biomarkers, parameters, representative evidence, and clinical outcomes, with full references and expanded abbreviations. (Pages 9-10)
- Risk-of-Bias assessment – Section 5 clarifies the rationale for using Cochrane RoB 2.0 for RCTs and Newcastle–Ottawa Scale (NOS) for cohort studies to ensure methodological transparency. (Page 10-12)
- Clinical vs. Conceptual Triad – Clarified that the regenerative triad represents a translational conceptual framework based on convergent mechanistic and clinical data; multicenter RCTs are planned to evaluate sequential use. (Pages3-15)
- Tables and abbreviations – All tables were reformatted for clarity, column spacing corrected, and full forms of all abbreviations included in legends and in the final abbreviation list. (Tables 1,2,3 Pages 9-13)
- Figures and visual elements –
Although no figures are included within the main manuscript, the graphical abstract has been thoroughly revised to enhance visual clarity, scientific precision, and alignment with the textual content. This comprehensive graphical abstract visually integrates the key mechanistic and translational concepts of the manuscript, thereby improving reader comprehension while avoiding redundancy with the comparative data presented in Tables 1–3. - Literature search method – Added a paragraph describing database (Page 2)
The literature cited in this perspective was systematically identified through searches of PubMed, Scopus, and Web of Science using the keywords “extracorporeal shockwave therapy,” “high-intensity laser therapy,” “sterile water injection,” and “chronic musculoskeletal pain.” The search was restricted to peer-reviewed publications between 2020 and 2025, including randomized controlled trials, cohort and comparative studies, mechanistic investigations, and systematic reviews or meta-analyses relevant to regenerative and rehabilitative medicine. Notably, the authors’ own previously published cohort studies (31,32), which investigated the comparative effectiveness and safety of ultrasound-guided mechanical needling with sterile-water injection for lumbar spinal stenosis and facet joint syndrome, respectively, were included as representative high-quality observational evidence. This comprehensive approach ensured that the synthesis reflects the most current and methodologically robust data supporting these non-pharmacological regenerative interventions.
11. Discussion and novelty – Expanded to emphasize mechanistic integration and translational novelty distinguishing the triad from traditional multimodal frameworks, highlighting its drug-free, mechanism-based design. (as all yellowed highlights in round 1)
12. Conclusion – Strengthened to highlight the clinical relevance, innovation, and public-health importance of the regenerative triad as a sustainable, opioid-sparing approach aligned with Healthy Ageing goals. (Pages 14-15)
13. Language quality – The entire manuscript underwent comprehensive English editing by a professional editor to enhance fluency, precision, and academic tone.

Reviewer 4 Report
Comments and Suggestions for Authors
I appreciate the authors' responses to most of the previous comments. To further improve methodological transparency, it I think it would be helpful to briefly clarify how the cited studies were identified and selected (research sources, inclusion criteria and time frame).
Author Response
Response to Reviewer 4
Comment 1:
I appreciate the authors' responses to most of the previous comments. To further improve methodological transparency, I think it would be helpful to briefly clarify how the cited studies were identified and selected (research sources, inclusion criteria, and time frame).
Response 1:
Thank you for this helpful suggestion. We have now added a clarification paragraph in the Methods Overview subsection of the manuscript.
Revised text (Page 2): This addition clarifies our literature selection strategy and strengthens methodological transparency.
The literature cited in this perspective was systematically identified through searches of PubMed, Scopus, and Web of Science using the keywords “extracorporeal shockwave therapy,” “high-intensity laser therapy,” “sterile water injection,” and “chronic musculoskeletal pain.” The search was restricted to peer-reviewed publications between 2020 and 2025, including randomized controlled trials, cohort and comparative studies, mechanistic investigations, and systematic reviews or meta-analyses relevant to regenerative and rehabilitative medicine. Notably, the authors’ own previously published cohort studies (31,32), which investigated the comparative effectiveness and safety of ultrasound-guided mechanical needling with sterile-water injection for lumbar spinal stenosis and facet joint syndrome, respectively, were included as representative high-quality observational evidence. This comprehensive approach ensured that the synthesis reflects the most current and methodologically robust data supporting these non-pharmacological regenerative interventions.

Round 3
Reviewer 3 Report
Comments and Suggestions for Authors
Biomedicines-3949084
Title: Regenerative and Drug-Free Strategies for Chronic Musculoskeletal
Pain: An Evidence-Based Perspective on Shockwave Therapy, High-Intensity
Laser Therapy, and Ultrasound-Guided Mechanical Needling with Sterile Water
Injection
Revised comments
- The author have revised the said manuscript. However,
- The author summarised all the response to several comments. However, while re-reviewing the manuscript, it became bit difficult to find the relevant information in the relevant section. For example, it is not clear if information about comment 2 ie whether the three modalities ESWT, HILT, SWI has been defined and if there are only 3 modalities available, also if the information of other relevant modalities has been mentioned with references in brief.
- Likewise, searching for other response is also not smooth. The Author must mention the comment number for which response has been provided for the smooth review of manuscript.
Comments on the Quality of English Language
English must be improved
Author Response
Response to Reviewer 3 (Round 3)
We sincerely thank the reviewer for the thoughtful and constructive feedback. We greatly appreciate the time and attention dedicated to the evaluation of our manuscript. In response to the comments provided, we have revised the manuscript and the response document to improve clarity, transparency, and ease of review, as detailed below.
Comment 1:
“The author summarised all the response to several comments. However, while re-reviewing the manuscript, it became bit difficult to find the relevant information in the relevant section… The Author must mention the comment number for which response has been provided for the smooth review of manuscript.”
Response 1:
Thank you very much for this helpful comment. We agree that clearer structuring improves navigation during re-review.
Therefore, we have revised the entire Response-to-Reviewers file so that each response is now explicitly labeled by comment number (e.g., Response to Comment 1, Response to Comment 2). This formatting ensures clear mapping between each reviewer comment and the corresponding revision.
Comment 2:
“…whether the three modalities ESWT, HILT, SWI have been defined and if there are only three modalities available…”
Response 2:
Thank you for highlighting this important point. We have now explicitly defined all three modalities of Extracorporeal Shockwave Therapy (ESWT), High-Intensity Laser Therapy (HILT), and Ultrasound-Guided Mechanical Needling with Sterile Water Injection (SWI) within the Introduction.
These definitions are accompanied by concise mechanistic explanations supported by recent PubMed-indexed references (2020–2025).
This revision is located on Page 2-9 references highlighted in red.
Comment 3:
“…if the information of other relevant modalities has been mentioned with references in brief.”
Response 3:
We appreciate this suggestion. To ensure contextual completeness, we have added a brief mention of Prolotherapy and other relevant regenerative, non-pharmacologic modalities, together with supporting citations.
This addition appears on Page 7, Section 4.5
Comment 4:
“Searching for other response is also not smooth.”
Response 4:
Thank you for this observation. To enhance transparency, we have ensured that:
• all newly added text and references are highlighted in red, and
• all prior revisions remain highlighted in yellow, as required.
This dual-color system provides complete traceability across the manuscript.
Final Clarification Note:
We have carefully reviewed the Round 3 comment and have incorporated all requested items including the modality definitions, the addition of other regenerative therapies, and the structured response formatting. If there is a specific remaining point the reviewer wishes us to refine, we would be grateful for guidance so that we may address it precisely according to the reviewer’s intention.
